# The Influence of COVID-19 on University Students’ Well-Being, Physical Activity, Body Composition, and Strength Endurance

**DOI:** 10.3390/ijerph192315680

**Published:** 2022-11-25

**Authors:** Robert Podstawski, Kevin John Finn, Krzysztof Borysławski, Aneta Anna Omelan, Anna Maria Podstawska, Andrzej Robert Skrzypczak, Andrzej Pomianowski

**Affiliations:** 1Department of Tourism, Recreation and Ecology, University of Warmia and Mazury in Olsztyn, 10-957 Olsztyn, Poland; 2Department of Nutrition, Kinesiology, and Health, University of Central Missouri, Warrensburg, MO 64093, USA; 3Institute of Health, The Angelus Silesius University of Applied Sciences, 58-300 Wałbrzych, Poland; 4Independent Researcher, 10-068 Olsztyn, Poland; 5Department of Internal Diseases with Clinic, University of Warmia and Mazury in Olsztyn, 10-719 Olsztyn, Poland

**Keywords:** university students, pandemic, strength-endurance abilities, anthropometric characteristics

## Abstract

Background: Very few scientific studies have simultaneously evaluated changes in well-being (WB), physical activity (PA), and strength endurance (SE) based on results from before and during the COVID-19 pandemic. Aim: The aim of the study was to assess WB, PA, and SE levels, as well as body composition parameters in university students before and during the pandemic. Methods: The study included 30 males and 30 females (mean age: 21.7 ± 2.51 and 21.6 ± 2.34 years, respectively). Well-being using the Self-Perception Questionnaire (POMS), PA from the Polish regular version of the International Physical Activity Questionnaire (IPAQ), body composition parameters using the InBody 270 analyzer, and SE based on the results of a 12 min test involving a Concept 2PM-5 rowing ergometer were assessed before and during the pandemic. Results: An assessment of university students’ WB revealed a significant increase in variables such as tension, depression, anger, confusion, and a significant decline in vigor (*p* < 0.05). In both sexes, PA and SE decreased significantly—from 2115.4 to 1822.8 METs-min/week and from 2184.5 to 2035 m, respectively, in males, and from 1793.5 to 1699.8 METs-min/week and from 2021.5 to 1943.8 m, respectively, in females. At the same time, body mass and BMI values increased significantly (*p* < 0.001) in both females and males. Conclusions: The COVID-19 pandemic caused a significant decrease in WB, PA and SE levels in young females and males, and led to a significant increase in their body mass and BMI.

## 1. Introduction

Coronavirus disease 2019 (COVID-19) is an infectious disease caused by the severe acute respiratory syndrome coronavirus 2 [1]. The novel COVID-19 coronavirus, or severe acute respiratory syndrome coronavirus-2 (SARS-CoV-2), has spread rapidly throughout the world since its emergence in December 2019. The virus has infected approximately 2.9 million people in more than 200 countries and caused more than 200,000 deaths at the time of writing [2]. Over 30 million people have been diagnosed with COVID-19, and more than 500,000 deaths have been attributed to COVID-19-related complications in the United States [3]. More than 28 million excess years of life were lost in 31 countries, with a higher rate in males than females. The excess years of life lost to COVID-19 in 2020 were more than five times higher than those associated with the seasonal influenza epidemic in 2015 [4]. According to Islam et al. [4], the highest excess years of life lost per 100,000 population were observed in Bulgaria (men: 6820 to 7710, 7260 on average, 95% confidence interval; women: 2740 to 4730, 3730 on average), Russia (men: 6550 to 7480, 7020 on average; women: 4530 to 4990, 4760 on average), Lithuania (men: 4750 to 6070, 5430 on average; women: 2310 to 2980, 2640 on average), the USA (men: 4170 to 4530, 4350 on average; women: 2320 to 2550, 2430 on average), Poland (men: 3540 to 4120, 3830 on average; women: 1630 to 2040, 1830 on average), and Hungary (men: 2490 to 3040, 2770 on average; women:, 1590 to 2240, 1920 on average).

Gender-specific differences between COVID-19 patients remain insufficiently investigated. However, adverse health effects associated with COVID-19 have been found to be more common in men than in women in all racial groups [3,5,6]. The number of confirmed infection cases has been growing rapidly, which has strained healthcare systems worldwide; therefore, the World Health Organization (WHO) and public health agencies have undertaken various measures to control the COVID-19 pandemic, such as self-isolation, social distancing, and lockdowns. Although strict sanitary policies are considered most effective in mitigating COVID-19 transmission, they contribute to weakening social relationships, potential loss of working and leisure hours, negative lifestyle changes, and psychological distress [7,8]. For this reason, the work environment was associated with an increased risk of exposure to COVID-19 and virus transmission [9,10]. Research has shown that essential workers were at greater risk of COVID-19 infection than nonessential workers [11,12]. Behavioral and non-pharmacological interventions, including the use of masks, self-isolation, quarantine, or even lockdown of entire territories and communities, were the strategies employed to prevent or at least curb disease transmission by physically separating people. Many community healthcare providers used the term “social distancing”, although “physical distancing” seems to be more appropriate [13,14]. Novel information and communication technologies (ICTs), including social networks and channels, enabled many individuals to keep in touch by using on-line meeting tools and videoconferencing solutions (“smart working”). Quarantine, i.e., the separation and restriction of movement of symptomatic persons and asymptomatic subjects who may have been exposed to a contagious disease [15], was effectively implemented during the severe acute respiratory syndrome (SARS) epidemic in mainland China in 2002–2003 [16].

Studies on quarantined patients revealed adverse psychological effects associated with isolation (anger, confusion, or even symptoms of post-traumatic stress) [17]. Due to strict home confinement, the COVID-19 crisis also exerted harmful psychological effects on athletes for a number of reasons. First, forced isolation causes physical distance from loved ones, which can cause significant frustration, compromise well-being (WB), or even contribute to mental disorders [18]. Individuals strive to satisfy their innate psychological needs, such as the need for authentic connections, physical closeness and love, which can be severely thwarted in an isolated or closed environment [19]. The resulting frustration can be exacerbated in physically active persons such as athletes [20]. In addition to affecting mental adjustment and sleep patterns, the daily stressors associated with the lack of basic physical activity—PA (including work)—during an ongoing pandemic penetrate deeply into family and personal life [21].

Similarly to other types of activities (international events, travel, etc.), sports events were cancelled and suspended in many countries. Even the Olympic Games 2020 (Tokyo, Japan) were postponed by 1 year. The maintenance of PA is a fundamental requirement for the general population, not only athletes. Physiological adaptation under the influence of physical training is a reversible process. In fact, most aspects of physiological adaptation are lost during prolonged periods of inactivity [22]. The rate of decline in different motor abilities varies, and degenerative changes occur more rapidly in endurance and strength abilities than speed and maximum strength abilities. Each week of physical inactivity decreases overall fitness levels by up to 10% [23].

College students were one of the most vulnerable social groups during the COVID-19 pandemic. Even before this global health crisis, university students had to face numerous challenges, including financial concerns, pressure to excel academically and participate in social activities, as well as mental health issues [24]. Academic campuses are places where students live, study, and enter into close social interactions. Universities are also lively cultural centers. The rapid spread of the COVID-19 epidemic significantly affected university life and led to profound changes in the work style of academic staff and students [25]. University students’ daily routines were undoubtedly changed, and their well-being and mental health were compromised during the pandemic. Concerns about their own health and the health of loved ones, changes in living circumstances, disruptions in sleep patterns and diet, constraints on social activities, academic challenges, and changes in coursework delivery were powerful stressors that increased self-reported levels of stress, anxiety, and depression [26,27]. Yon et al. [28] described the change in self-reported PA and psychological mindsets of 216 college students in the southeastern region of the United States. They relied on the Intuitive Exercise Scale to determine the most prominent dimensions of exercise during the pandemic, and they found that exercising for fun, exercising to influence emotions, and loss of motivation to exercise were the emerging themes. Yu et al. [29] tested the relationship between PA and fitness in 1505 university students after one year of the COVID-19 lockdown using the International Physical Activity Questionnaire (Chinese version) and selected fitness measures according to gender. A comparison of the measures taken before (annually for three years) and during the COVID-19 pandemic (fourth year) revealed a significant decline in PA levels in females and a decline in overall physical fitness in both males and females. Similarly to the studies conducted on the general population, research into college students’ physical and sports activity during the pandemic revealed an overall decline in the frequency, duration, and intensity of exercise [30,31,32].

The pandemic’s simultaneous impact on university students’ well-being (WB), PA levels, body composition, and specific motor abilities such as strength endurance (SE) need to be investigated. Therefore, the aim of this study was to assess the WB, PA, body composition parameters, and SE of university students before and during the COVID-19 pandemic.

## 2. Materials and Methods

### 2.1. Participants

The study involved full-time students of the Department of Tourism and Recreation at the University of Warmia and Mazury in in Olsztyn (UWM). In the first stage of the study, which took place just before the outbreak of the COVID-19 pandemic (from 2 to 6 December 2019), the students participated in regular PA, including scheduled activities such as soccer, tennis, and volleyball at 90 min per week for each activity (total of 270 min/week). The students also pursued other types of PA such as recreational games and hiking. The second stage of the study was performed during the pandemic (from 7 to 11 June 2021) when all activities (except direct-contact PA) were conducted remotely. As part of the lockdown measures implemented by the Polish government, all citizens were required to remain indoors and were allowed to leave their homes only to shop for essential food items or walk their dogs. Pursuant to a decision of the Rector of the UWM in Olsztyn, students enrolled in physiology classes as well as medicine and veterinary medicine students were among the few groups who had to participate in “direct contact” PE classes on campus. The study participants had to comply with pandemic-related restrictions and test negative for SARS-CoV-2 in the RT-PCR test. All study participants took an RT-PCR test on 4 June. During the study, participants were also regularly checked for symptoms of COVID-19 infection based on the guidelines of the National Sanitary Inspectorate.

The potential participants were informed about the purpose of the study. The students who agreed to participate in the study (41 women and 37 men) were notified by e-mail and text message whether they met the inclusion criteria, and were provided with the date of final recruitment. All applicants were asked to complete a health questionnaire before the study. Ultimately, 30 women aged 20–23 years (mean age 21.6 ± 2.34 years) and 30 men aged 20–24 years (mean age 21.7 ± 2.51 years) who met the inclusion criteria (see below) were admitted to the study based on a medical examination. The participants confirmed that they did not take any medications or nutritional supplements, were in good health, and had no history of blood diseases or diseases affecting biochemical and biomechanical factors. None of the evaluated participants had respiratory or circulatory ailments. None of the subjects had symptoms of COVID-19 infection described in the National Sanitary Inspection Service guidelines. Based on their age, lifestyle, and field of study, male and female participants were regarded as homogeneous and representative of the examined population.

### 2.2. Ethical Approval

The study (Table 1) was conducted upon the prior consent of the Ethics Committee of the University of Warmia and Mazury in Olsztyn (No. 39/2011), Poland. The participants were volunteers who signed an informed consent statement.

### 2.3. Instruments and Procedures

#### 2.3.1. Assessment of Well-Being

Well-being was assessed with the Profile of Mood States (POMS) questionnaire. The questionnaire consists of 65 mood adjectives and measures six different mood states: tension, depression, anger, fatigue, confusion, and vigor. The “tension” subscale describes muscle tension and generalized discomfort. The “depression” subscale describes a mood of sadness with a sense of inadequacy and worthlessness. “Anger” is described as a mood of irritability and hostility toward others. “Vigor” is a subscale that describes readiness for action and invigoration. “Fatigue” is defined as a feeling of fatigue and low energy levels. In contrast, the “Confusion” subscale describes a state of confusion and a sense of ineffectiveness. The intensity of the feelings experienced at a given moment is described on a 5-point scale: 0—definitely not, 1—rather not, 2—hard to say, 3—rather yes, 4—definitely yes. The POMS questionnaire is a valid and reliable tool characterized by overall detection rate 80% with a sensitivity of 55% and a specificity of 84% [33], for measuring negative mood states, such as psychological distress, as well as positive mood states, such as vigor [34], including in conjunction with PA [35].

#### 2.3.2. Assessment of Physical Activity Levels

The students’ PA levels were determined using the short Polish version of the standardized and validated International Physical Activity Questionnaire (IPAQ) [36,37]. The IPAQ results were used to select a homogenous sample of male and female participants, and they were expressed in Metabolic Equivalent of Task (MET) units to present the PA levels of students. The participants were asked to estimate the average duration (minutes) of their weekly PA (minimum of 10 min) before the study. Based on the frequency, intensity, and duration of PA declared by the students, they were divided into groups representing low (<600 METs-min/week), moderate (600 to 1500 METs-min/week), and high (>1500 METs-min/week) PA levels.

#### 2.3.3. Body Composition Parameters

Body height was measured to the nearest 1 mm with a calibrated Soehlne Electronic Height Rod 5003 (Soehlne Professional, Backnang, Germany) connected to the InBody 720 body composition analyzer. Body composition parameters, including body mass, total body water (TBW), protein and mineral content, body fat mass (BFM), fat-free mass (FFM), skeletal muscle mass (SMM), percent body fat (PBF), body mass index (BMI) waist-hip ratio (WHR), InBody score, target weight, basal metabolic rate (BMR), and degree of obesity, were calculated by InBody 720 software [38].

#### 2.3.4. Assessment of Strength Endurance in a 12 min Rowing Ergometer Test

Strength endurance was assessed in a 12 min rowing ergometer test. The test’s accuracy and reliability had been validated previously [39,40]. In other studies, the rowing ergometer test was regularly applied to assess SE levels in female and male university students [41,42]. The test involved a Concept 2 PM5 standardized rowing ergometer (PH Markus, Szczecin, Poland) which is widely used to measure the SE of athletes [43] and university students [41,42]. The subjects participated in a general 10 min warm-up routine before the test [44]. The distance and average power measured during the test were manually entered into an Excel database in the ErgDate application (https://www.concept2.com/service/software/ergdata) accessed on 21 January 2019 via a Samsung Galaxy A53 5G smartphone (Samsung Electronics, Warsaw, Poland) for future analyses.

The assessed students (men and women) attended exercise physiology classes in the first stage of the study (winter semester) and anthropometry classes in the second stage of the study (summer semester). Before the first stage, the participants received detailed information about the purpose of the study, potential risks, measurement methods, and the technique for performing the rowing ergometer test. The relevant information was provided during physiology classes and lectures before the study as part of the standard curriculum for first-year tourism and recreation students. In the second stage, the relevant information was provided during anthropometry lectures conducted online on the Teams platform. The second stage of the study was conducted during the lockdown (second half of April), when physiology classes were compressed into a single week and the students stayed on campus. The correct technique for rowing on an ergometer was communicated by e-mail via the university’s USOS Web system. The students were provided with links to instructional videos and online rowing exercise programs. The students were also able to practice their paddling technique during three classes immediately preceding the study. The row stroke is divided into two phases (Figure 1). Phase I (pulling the bar): from the starting position with bent legs, straight arms and a forward leaning back (a), the lower limbs are gradually straightened (b); the torso is abducted backward to a maximum of 45° (c), and the bar is pulled to the chest (d). Phase II (bar inversion): from a position with straight legs, abducted and pulled back torso (d), the arms are extended, the torso is bent forward (c), and the lower limbs are flexed to a position with maximally bent legs, forward bent torso and straight arms (a).

The participants were not required to wear a mask when performing motor tests on the rowing ergometer because all of them had tested negative in the RT-PCR test. The use of masks is generally not recommended during such activities, especially strenuous exercise.

### 2.4. Statistical Analysis

Anthropometric, body composition, physiological, and psychological (POMS) parameters were processed with the use of descriptive statistics in the Statistica PL v. 13.5 program. The arithmetic means of all parameters measured in both stages of the study were compared in the Student’s *t*-test for dependent samples. All parameters exhibited normal distribution. Normality was verified by the Shapiro–Wilk test. The asymmetry coefficient As (skewness) was also calculated. Values within the 95% confidence interval were regarded as statistically significant (*p* < 0.05).

## 3. Results

### 3.1. Analysis I: Assessment of Well-Being

The WB of male and female respondents before and during the pandemic is compared in Table 2. In both genders, well-being variables, including tension, depression, anger, and confusion, increased significantly during the pandemic (*p* < 0.001 for all variables, excluding anger in women, where *p* = 0.050). In contrast, vigor was significantly higher in both sexes before the pandemic (*p* < 0.001). No significant (*p* > 0.05) differences in fatigue levels were observed between the compared periods (Table 2) in male or female students.

The WB of male and female students before and during the pandemic is compared in Table 3. No significant differences (*p* < 0.05) in WB parameters (tension, depression, anger, fatigue, confusion, and vigor) were observed between men and women before the pandemic. In contrast, during the pandemic, variables such as fatigue and vigor increased significantly in men (*p* = 0.034 and *p* = 0.007, respectively).

### 3.2. Analysis II: Body Composition Parameters

Body composition parameters in men and women before and during the pandemic are compared in Table 4. In male and female students, a significant increase in body mass (1.6 and 1.1 kg, respectively, *p* < 0.001), BMI (0.5 and 0.4 kg/m^2^, respectively, *p* < 0.001), and weight control (1.6 and 1.4 kg, respectively, *p* < 0.05) was noted during the COVID-19 pandemic. No significant differences (*p* > 0.05) in the remaining body composition parameters were observed between the compared periods.

### 3.3. Analysis III: Assessment of Strength Endurance

Descriptive statistics and differences in the arithmetic means of physiological parameters in men and women before and during the pandemic are presented in Table 5. PA levels were significantly (*p* < 0.001 and *p* < 0.05, respectively) higher in male and female subjects before than during the pandemic, although this parameter was high (>1500 METs-min/week) in both periods. Strength endurance levels (expressed by the distance in meters traveled on a rowing ergometer in 12 min) were also significantly (*p* < 0.001) higher before than during the pandemic. PA levels in men and women decreased by 292.6 and 93.7 METs-min/week, respectively, during the pandemic, whereas the distance traveled decreased by 149.5 and 77.7 m, respectively. Accordingly, exercise characteristics such as power (men: decrease from 87.2 to 68.8 watts, difference—18.4 watts, *p* = 0.003; women: decrease from 66.9 to 57.5 watts, difference—9 watts, *p* < 0.001), energy expenditure (men: decrease from 118.0 to 105.9 kcal, *p* = 0.003; women: non-significant decrease, *p* > 0.05) also decreased significantly during the pandemic. Mean and maximum HR values also decreased significantly (HR_avg_ decreased by 8.6 bpm, and HR_max_ decreased by 3.4 bpm in both groups; *p* < 0.001). Exercise times in specific intensity ranges (from easy to maximum) also decreased significantly during the pandemic. In both men and women, a significant decrease (*p* < 0.001) in exercise time was observed in the easy range (men: 81.0 sec before the pandemic, 110.4 s during the pandemic; women: 86.0 s before the pandemic, 105.0 s during the pandemic), as well as the moderate range (men: 187.6 s before the pandemic, 207.2 s during the pandemic; women: 143.8 s before the pandemic, 172.4 s during the pandemic). In contrast, very difficult and maximal efforts in both sexes lasted significantly longer (*p* < 0.017) before than during the pandemic (men: before—161.1 and 66.4 s before the pandemic, 150.0 and 69.6 s during the pandemic, respectively; women: 164.5 and 106.2 s before the pandemic, 116.1 and 113.9 s during the pandemic, respectively), whereas the ranges of difficult efforts did not differ significantly (Table 5).

## 4. Discussion

In the future, comprehensive analyses of WB, PA, body composition, and SE could play an important role in the decision-making process to protect students against the negative consequences of the COVID-19 pandemic. The results of the present study clearly indicate that the strategies and measures implemented during the pandemic to limit the transmission of the virus significantly contributed to a decline in university students’ WB, PA levels, SE, and body composition parameters, in particular by increasing their body mass and BMI. In comparison, Chwalczynska and Andrzejewski [45] reported significant changes in body composition between December 2019 (four months before the lockdown) and February 2021 (the third wave of the pandemic). In the cited study, body mass and the body mass index increased in male students, whereas fat mass increased in female students.

The results of published studies [26,28] as well as the responses of the participants in the present study indicate that many college students had to cope with new stressors during the pandemic, which exerted a negative impact on their emotions, mental health, and PA. Regarding the students’ WB, it should be noted that both male and female participants were characterized by high PA levels not only before, but also during the pandemic. The above could be attributed to their academic profile and obligatory participation in various types of PA. According to some studies, Polish students enrolled in health-related university programs (such as physiotherapy, physical education, tourism and recreation) had declared high levels of PA (2000–10,000 METs-min/week) before the lockdown [46,47]. However, the results of our study indicate that the COVID-19 pandemic had a negative impact on the surveyed students’ PA levels. Chinese students also declared that their PA levels had decreased during the pandemic, and more than 50% of the surveyed respondents did not meet the PA guidelines of the WHO [48]. A study of Italian university students revealed a significant decrease in vigorous PA, moderate PA, and, especially, walking time during the pandemic [49]. However, it should be noted that in Poland and in other countries, sports and recreational facilities (swimming pools, gyms, fitness clubs, playgrounds) were closed, and organized physical activities were cancelled during the lockdown. Outdoor sports and recreation opportunities were also limited [50]. Thus, it can be concluded that the lockdown was one of the key reasons for the decrease in the PA levels of the surveyed students. However, it is also possible that the students’ motivation to exercise had not been very strong or well established before the pandemic, and students found it difficult to maintain pre-pandemic PA levels in a crisis situation.

Individuals who participate in compulsory physical activities and activities organized by others generally find it easier to remain physically active. During the pandemic, sports facilities were closed down, and team and club activities were suspended in Poland and in other countries [51]. As a result, limited training opportunities were a source of considerable stress for physically active individuals. The absence of organized training sessions was an additional stressor for athletes, and a similar phenomenon also occurred among tourism and physical recreation students who participate in sports activities, which may have been because they were characterized by high levels of physical activity [52]. The closure of gyms and fitness classes also decreased the PA of university students [31,53]. Athletes who were prevented from training almost overnight also experienced considerable stress [54]. The severity of stress can differ among individuals who practice various sports and physical activities. According to Makarowski et al. [55], martial arts athletes experience less stress than other athletes. It should also be noted that PA delivers beneficial effects by protecting the body against the negative impact of viral infections. PA may influence immune responses and the effectiveness of defense mechanisms [56,57,58]. There is strong evidence to indicate that high-endurance sports such as running, cycling, rowing, and swimming significantly increase B cell, T cell, neutrophil, and NK cell counts in the circulatory system [59,60]. Participation in aerobic and muscle-strengthening activities above the minimum recommended time provides additional health benefits and results in higher levels of SE [61], including endurance-strength abilities (SE) [62,63]. Therefore, a significant decrease in the PA levels of the examined university students led to a significant decrease in SE in both sexes [64]. There is mounting evidence to suggest that activities which increase muscular strength and endurance deliver health benefits in non-elderly populations [65,66]. For example, resistance exercises that increase mechanical loading on skeletal tissues can effectively stimulate bone formation in young adults and slow bone loss in middle-aged individuals [67].

The analysis of exercise indicators revealed a significant decrease in the rowing performance of the examined students. The power developed in the 12 min rowing ergometer test and, consequently, the traveled distance decreased during the pandemic. Before the pandemic, maximal and mean HR values were significantly higher during the test, and exercise times in the very difficult and maximal effort ranges were significantly longer in both sexes. Regarding endurance and strength abilities, the guidelines for promoting and maintaining good health and physical independence state that adults should perform exercises that maintain or increase muscular strength and endurance minimum twice a week. These recommendations include vigorous-intensity PA which is more effective in promoting SE [61,68]. The decrease in SE scores during the pandemic can be largely attributed to significantly lower PA levels and changes in the POMS profile.

### Strengths and Limitations

The strength of this research is the multivariate analysis of changes in WB, PA, body composition parameters, and SE during the COVID-19 pandemic, which involved a representative and homogenous group of students with high PA levels. The present study relied on a holistic approach by examining changes not only in the participants’ physiological parameters, but also in their mental state.

One of the limitations of the present study is that the daily routines of university students were not monitored during the time spent at home or away from the university. It was assumed that restricted access to university facilities during the pandemic affected the participants’ PA patterns. The use of an objective device (such as a PA tracker) for monitoring student behaviors would greatly improve the validity of PA measurements and would provide additional data for identifying the underlying causes of the decrease in PA levels, SE, and WB. Another limitation is the lack of a comparative reference group for analyzing changes in body composition over the examined 18-month period (December 2019–June 2021). Weight gain and changes in body composition are common among university students even without pandemic-related restrictions. In first-year students, changes in body composition result from changes in eating patterns, as students migrate to “independent living” on campus. A daily dietary recall might have been helpful to deduce if changes in body composition are linked to something other than PA.

## 5. Conclusions

The results of the study indicate that the COVID-19 pandemic negatively affected university students’ WB and led to a significant decrease in their PA and SE levels, as well as a significant increase in body mass and BMI. A similar pattern of changes was observed in both male and female students. The observed decline in the mood of university students surveyed during the pandemic is not surprising. The pandemic paralyzed the world, and it came as a shock to most people who were concerned about their health and lives, as confined to their home, and forced to live in social isolation. The need for freedom, social contact, and entertainment was particularly prominent in youths and young adults (including university students) craved for PA to relieve the mental and physical stress caused by forced isolation. The relationship between PA and well-being has been studied and reported in the scientific literature, indicating that students would have benefitted, both physically and mentally, if at least some physical activities were accessible during the pandemic (upon the observance of strict sanitary measures). The problems associated with the pandemic has not yet been completely solved, and the risk of serious complications or even death still exists, including among individuals with asymptomatic infections. Individuals can minimize these risks by searching for health information and paying attention to their health [69]. The opportunities to safely engage in physical activities during similar crises in the future need attention. The authorities should also support measures to foster sports and recreational habits among young people. Such measures would, at least partly, decrease the prevalence and severity of health problems (physical and mental) during lockdowns which could still happen as COVID continues to evolve.

## Figures and Tables

**Figure 1 ijerph-19-15680-f001:**
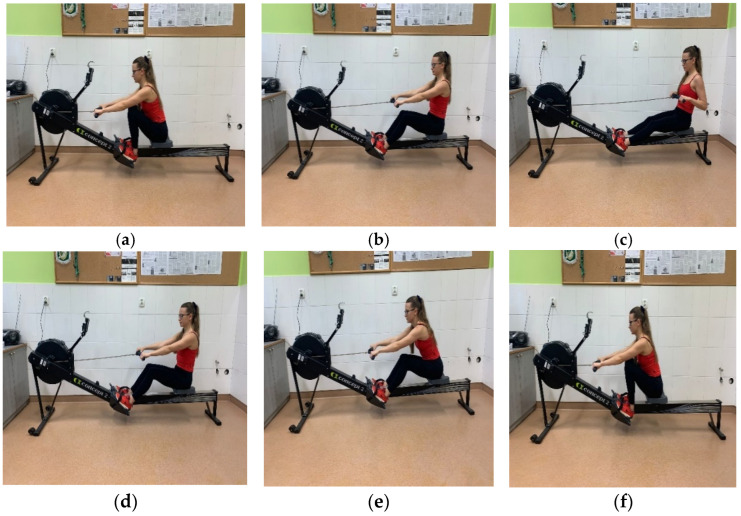
Rowing phases (**a**–**f**) on the Concept 2 PM5 indoor rowing ergometer.

**Table 1 ijerph-19-15680-t001:** Age and body height of male and female university students at the beginning of the study.

Variables	Mean	SD	Min–Max
Age of men [years]	19.40	0.89	18–21
Age of women [years]	19.13	0.82	18–20
Body height of men [cm]	180.67	6.41	170–196
Body height of women [cm]	166.57	4.72	158–178

Notes: SD—standard deviation, Min—minimum value, Max—maximum value.

**Table 2 ijerph-19-15680-t002:** Descriptive statistics and differences in the mean values of the POMS parameters in men (N = 30) and women (N = 30) before and during the COVID-19 pandemic.

POMS Variables	Before Pandemic				During Pandemic				Differences	
	Mean	SD	Min–Max	As	Mean	SD	Min–Max	As	*t*	*p*
Men										
Tension	0.65	0.38	0.00–1.56	0.88	1.33	0.49	0.56–2.56	0.80	7.79	<0.001 *
Depression	0.22	0.37	0.00–1.33	1.61	0.80	0.52	0.07–2.73	1.74	7.97	<0.001 *
Anger	0.71	0.25	0.25–1.42	1.03	1.15	0.46	0.34–2.75	1.43	6.75	<0.001 *
Fatigue	1.09	0.81	0.00–2.57	0.54	1.33	0.59	0.29–2.43	0.08	−1.44	ns
Confusion	0.90	0.49	0.29–2.14	1.24	1.37	0.61	0.14–3.29	1.11	5.19	<0.001 *
Vigor	2.88	0.53	1.63–3.63	−0.61	1.97	0.52	0.75–2.88	−0.20	−8.30	<0.001 *
Women										
Tension	0.63	0.46	0.00–2.00	1.38	1.39	0.49	0.67–2.67	0.80	12.54	<0.001 *
Depression	0.26	0.41	0.00–1.60	1.85	0.97	0.45	0.20–2.13	0.41	10.67	<0.001 *
Anger	0.77	0.56	0.25–3.25	1.68	0.99	0.50	0.08–1.92	0.18	2.00	0.050
Fatigue	1.25	0.72	0.14–2.57	0.14	1.03	0.47	0.14–2.00	0.32	−1.78	ns
Confusion	0.84	0.44	0.00–1.86	0.33	1.30	0.51	0.57–2.57	0.49	6.56	<0.001 *
Vigor	2.83	0.58	1.75–3.75	−0.61	1.62	0.44	0.75–2.50	−0.22	−13.06	<0.001 *

Notes: SD—standard deviation, As—asymmetry coefficient, ns—not significant, *t*—value of *t*-function, *p*—significance level, * *p* < 0.001.

**Table 3 ijerph-19-15680-t003:** Sexual dimorphism of POMS parameters before and during the COVID-19 pandemic.

POMS Variables	Before Pandemic						During Pandemic					
	Men		Women		Difference (M-W)		Men		Women		Difference (M-W)	
	Mean	SD	Mean	SD	*t*	*p*	Mean	SD	Mean	SD	*t*	*p*
Tension	0.65	0.38	0.63	0.46	0.20	ns	1.33	0.49	1.39	0.49	−0.44	ns
Depression	0.22	0.37	0.26	0.41	−0.42	ns	0.80	0.52	0.97	0.45	−1.39	ns
Anger	0.71	0.25	0.77	0.56	−0.54	ns	1.15	0.46	0.99	0.50	1.29	ns
Fatigue	1.09	0.81	1.25	0.72	−0.82	ns	1.33	0.59	1.03	0.47	2.17	0.034
Confusion	0.90	0.49	0.84	0.44	0.47	ns	1.37	0.61	1.30	0.51	0.42	ns
Vigor	2.88	0.53	2.83	0.58	0.38	ns	1.97	0.52	1.62	0.44	2.81	0.007

Notes: M—men, W—women, *t*—value of *t*-function, *p*—significance level, ns—not significant.

**Table 4 ijerph-19-15680-t004:** Differences in the mean values of body composition parameters in men (N = 30) and women (N = 30) before and during the COVID-19 pandemic.

Variables	Before Pandemic (BP)				During Pandemic (DP)				Difference (DP-BP)	
	Mean	SD	Min–Max	As	Mean	SD	Min–Max	As	*t*	*p*
Men										
TBW [kg]	46.25	5.95	31.6–58.2	−0.11	46.65	5.96	34.9–61.0	0.48	0.64	ns
Proteins [kg]	12.54	1.61	8.6–15.8	−0.07	12.65	1.62	9.4–16.5	0.47	0.68	ns
Minerals kg]	4.32	0.60	2.9–5.7	0.05	4.39	0.65	3.07–6.09	0.66	0.94	ns
Body mass [kg]	75.10	11.13	50.5–108.9	0.83	76.66	10.74	54.3–110.3	0.94	4.86	<0.001 *
BMI [kg/m^2^]	22.98	2.93	17.3–29.1	0.56	23.45	2.75	18.6–29.0	0.68	4.81	<0.001 *
BFM [kg]	11.98	6.01	4.0–29.2	1.41	12.96	6.02	6.7–28.4	1.51	1.12	ns
FFM [kg]	63.12	8.14	43.1–79.7	−0.10	63.70	8.22	47.4–83.3	0.49	0.67	ns
SMM [kg]	35.88	4.89	23.7–45.8	−0.11	36.20	4.91	26.5–47.7	0.48	0.61	ns
PBF [%]	15.57	6.02	6.1–29.8	0.69	16.59	5.99	9.5–30.3	0.94	0.93	ns
In Body Score	78.0	6.9	65–91	−0.05	79.5	7.5	65–93	−0.08	1.01	ns
Weight control [kg]	0.06	6.88	−15.7–15.4	−0.20	−1.57	5.89	−16.6–9.6	−0.99	−2.04	0.05
BMR [Kcal]	1733.4	175.8	1300–2091	−0.10	1745.7	177.6	1395–2170	0.49	0.66	ns
WHR	0.85	0.07	0.7–1.0	0.82	0.85	0.06	0.7–1.0	0.98	0.12	ns
Women										
TBW [kg]	32.24	3.19	26.9–40.7	0.93	32.3	3.37	26.0–41.9	0.56	0.11	ns
Proteins [kg]	8.64	0.87	7.2–10.9	0.85	8.68	0.91	7.0–11.2	0.49	0.18	ns
Minerals kg]	3.18	0.38	2.6–4.1	0.86	3.18	0.32	2.5–4.0	0.31	0.02	ns
Body mass [kg]	60.41	7.90	47.6–83.7	0.96	61.51	8.14	48.4–85.6	1.07	3.93	<0.001 *
BMI [kg/m^2^]	21.74	2.42	17.5–29.7	1.21	22.14	2.48	17.8–30.3	1.24	3.95	<0.001 *
BFM [kg]	16.35	5.73	5.3–29.3	0.24	17.35	6.20	8.9–41.2	1.67	1.18	ns
FFM [kg]	44.07	4.43	36.7–55.7	0.92	44.16	4.59	35.5–57.1	0.54	0.11	ns
SMM [kg]	24.11	2.65	19.7–31.2	0.96	24.17	2.76	18.9–31.9	0.52	0.13	ns
PBF [%]	26.52	6.77	9.7–36.1	−0.74	27.72	6.38	16.3–48.1	0.89	0.90	ns
In Body Score	73.5	4.7	67–85	0.57	73.9	6.0	54–85	−0.94	0.29	ns
Weight control [kg]	0.08	5.92	−13.1–12.4	−0.12	−1.45	7.05	−27.9–10.1	−1.83	−2.32	0.027
BMR [Kcal]	1322.1	95.6	1163–1573	0.92	1323.9	99.3	1137–1604	0.54	0.10	ns
WHR	0.85	0.05	0.8–1.0	0.55	0.85	0.04	0.8–0.9	0.59	0.02	ns

Notes: TBW—total body water, BMI—body mass index, BFM—body fat mass, FFM—fat-free mass, SMM—skeletal muscle mass, PBF—percent body fat, BMR—basal metabolic rate, WHR—waist-hip ratio, * *p* < 0.001, ns—not significant.

**Table 5 ijerph-19-15680-t005:** Differences in the mean values of physiological parameters in men (N = 30) and women (N = 30) before and during the COVID-19 pandemic.

Parameter	Before Pandemic				During Pandemic				Difference	
	Mean	SD	Min–Max	As	Mean	SD	Min–Max	As	*t*	*p*
Men										
PA level [METs-min/week]	2115.4	768.7	720–3545	−0.14	1822.8	635.8	705–2964	−0.19	−7.10	<0.001 *
Distance [m]	2184.5	423.8	1577–3017	0.33	2035.0	346.7	1563–2745	0.29	−3.82	<0.001 *
Power [W]	87.2	49.8	29–206	0.75	68.8	34.5	29–155	0.74	−3.24	0.003
Energy expenditure [Kcal]	118.0	34.2	78–200	0.85	105.9	24.0	78–165	0.71	−3.27	0.003
SPM [swing/min]	25.7	2.9	20–32	0.04	25.4	3.1	19–32	−0.44	−0.69	ns
HR_avg_ [bpm]	140.3	18.6	105–170	−0.02	131.7	13.8	107–155	−0.16	−5.98	<0.001 *
HR_max_ [bpm]	177.1	16.7	146–199	−0.57	173.7	17.1	142–196	−0.66	−7.14	<0.001 *
Exercise time in intensity ranges [s]										
Easy < 107 [bpm]	81.0	84.5	0–359	1.46	110.4	87.1	0–399	1.27	5.41	<0.001 *
Moderate 107–124 [bpm]	187.6	135.3	12–462	0.41	207.2	137.0	1–471	0.44	3.43	0.002
Difficult 125–141 [bpm]	223.9	126.3	24–523	0.59	216.7	121.8	38–493	0.67	−0.90	ns
Very Difficult 142–159 [bpm]	161.1	151.5	1–427	0.69	141.4	150.0	1–481	0.97	−2.52	0.017
Maximal ≥ 160 [bpm]	66.4	95.3	0–323	1.52	44.3	69.6	0–263	1.95	−3.35	0.002
Women										
PA [METs-min/week]	1793.5	663.3	730–2842	−0.23	1699.8	582.8	590–2640	−0.43	−2.05	0.05
Distance [m]	2021.5	274.9	1589–2551	0.40	1943.8	241.4	1395–2434	−0.23	−3.81	<0.001 *
Power [W]	66.9	27.2	30–125	0.72	57.5	20.3	20–108	0.48	−3.54	<0.001 *
Energy expenditure [Kcal]	101.2	23.8	22–144	−0.76	98.0	14.0	72–132	0.37	−1.26	ns
SPM [swing/min]	25.1	4.6	17–36	0.52	25.0	4.3	19–34	0.37	−0.09	ns
HR_avg_ [bpm]	138.9	18.3	105–182	0.48	134.7	14.2	104–170	0.30	−4.28	<0.001 *
HR_max_ [bpm]	178.7	16.8	146–199	−0.96	175.1	16.4	142–198	−0.95	−9.09	<0.001 *
Exercise time in intensity ranges [s]										
Easy < 107 [bpm]	86.0	67.0	0–224	0.33	105.0	71.8	0–231	0.10	5.14	<0.001 *
Moderate 107–124 [bpm]	143.8	95.8	14–279	−0.26	172.4	103.7	21–332	–0.28	6.91	<0.001 *
Difficult 125–141 [bpm]	219.6	76.1	71–348	−0.25	223.4	59.1	116–348	0.38	0.40	ns
Very Difficult 142–159 [bpm]	164.5	124.7	3–523	0.91	142.6	116.1	2–442	0.52	−4.32	<0.001 *
Maximal ≥ 160 [bpm]	106.2	136.1	0–438	1.32	76.7	113.9	0–405	1.56	−4.34	<0.001 *

Notes: PA—physical activity, SPM—strokes per minute, HR—heart rate, * *p* < 0.001, ns—not significant.

## Data Availability

The access to research data has been restricted by the Ethics Committee of the University of Warmia and Mazury in Olsztyn (UWM), Poland, to protect the participants’ privacy. The data can be released to researchers who meet the criteria for access to confidential data upon reasonable request (podstawskirobert@gmail.com).

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
