# Peer review of "The Influence of COVID-19 on University Students’ Well-Being, Physical Activity, Body Composition, and Strength Endurance"

_ijerph, 2022, doi:10.3390/ijerph192315680_

Round 1

Reviewer 1 Report

Dear authors,

Thank you very much for giving me the opportunity to review your study. It is very interesting and innovative.
Here are my comments which I hope will help you to improve it.
I recommend not to use abbreviations in the abstract.
In the keywords you use women and men, perhaps if you replace them with gender it would be enough.
The introduction is very interesting, but there are paragraphs that have nothing to do with the study itself. For example, when it talks about essential and non-essential workers. The study is really about university students and the consequences for them are hardly mentioned in the introduction.
There is no indication in the methodology of the exact time period over which the assessments are made. This should be included.
The participation rate of students is not known either.
Measuring instruments that are scales should include reliability data.
The rowing test is understood to be face-to-face. How could students come during the pandemic to take the test if there was confinement? This point should be made clear.
Abbreviations appearing in a table should be explained at the end of the table.
Have you considered doing any regression analysis? Statistical analyses are too basic.
The discussion talks about students as if they were competitive athletes and I understand that this is not the case. Also, the discussion is confusing and short.
Strengths are not strengths. Just because there are many variables does not mean that the statistical analysis is adequate. Holistic health includes the social domain as well.
The conclusions are too brief.

The reviewer.

Author Response

Dear Reviewer,

You will find our response in attachment.

Best regards,

Robert Podstawski

International journal of EnvironmentaL Research and Public Health

14 November 2022

            Below you will find our responses to the review of the article entitled “The influence of COVID-19 on university students’ well-being, physical activity, body composition, and strength endurance”. Manuscript ID: ijerph-2004320

Above all, we would like to thank the Reviewers for their valuable contributions to preparing the paper for publication. The Reviewers’ comments have been taken into consideration in the revision process, and changes in the manuscript are highlighted in yellow. A detailed description of the changes is presented below in Table 1.

We would like to thank both Reviewers for extensive and thorough reviews which have helped us to improve the quality of this manuscript. We hope that the revised manuscript is suitable for publication. We are always open to constructive criticisms, and should the Reviewers have any additional comments, we ask that they provide us with specific suggestions. Once again, thank you for the time devoted to reviewing our manuscript.

Yours sincerely,

Robert Podstawski, on behalf of all co-authors.

Table 1. Specific comments and responses

Comment

Response

Responses to the comments made by Reviewer 1

Dear authors,

Thank you very much for giving me the opportunity to review your study. It is very interesting and innovative. Here are my comments which I hope will help you to improve it.

Thank you very much for the positive feedback and for recognizing our study as innovative and interesting.

I recommend not to use abbreviations in the abstract.

Thank you very much for this comment. In the current version, the abstract has 234 words and the limit is 250 words. Therefore, abbreviations (spelled out at first mention) were used in the abstract to reduce the word count, which is common practice. Please note that abbreviations such as BMI, IPAQ, POMS and COVID-19 are widely used in scientific publications; therefore, we would like to keep them in the abstract.

In the keywords you use women and men, perhaps if you replace them with gender it would be enough.

The words “women” and “men” were removed from the keywords.

The introduction is very interesting, but there are paragraphs that have nothing to do with the study itself. For example, when it talks about essential and non-essential workers. The study is really about university students and the consequences for them are hardly mentioned in the introduction.

In line with the Reviewer's comment, additional information about university students was provided in the Introduction section. However, we would like to stress that this type of research has been rarely conducted among university students. Furthermore, we wanted to accurately describe the global situation during the pandemic, which is why the distinction between essential and non-essential workers was made.

There is no indication in the methodology of the exact time period over which the assessments are made. This should be included. The participation rate of students is not known either.

The study period was described in the Materials and Methods section.

Measuring instruments that are scales should include reliability data.

Please note that the reliability of all research tools has been confirmed by the relevant literature. A new reference item was included for the IPAQ questionnaire: Craig, C. L.; Marshall, A. L.; Sjöström M.; Bauman, A. E.; Booth, M. L.; Ainsworth, B. E.; et al. International Physical Activity Questionnaire: 12-country reliability and validity. Med Sci Sports Exerc. 2003, 35, 1381–1395.

The following reference was added to confirm the reliability of the Concept-2 rowing ergometer: Treff, G.; Mentz, L.; Mayer, B.; Winkert, K.; Engleder, T.; Steinacker, J.M. Initial evaluation of the Concept-2 rowing ergometer’s accuracy using a motorized test rig. Front Sports Act Living 2022, 3, 801617. doi: 10.3389/fspor.2021.801617.

The rowing test is understood to be face-to-face. How could students come during the pandemic to take the test if there was confinement? This point should be made clear.

The relevant explanation was provided in the Materials and Methods section.

Abbreviations appearing in a table should be explained at the end of the table.

Abbreviations were explained under the tables.

Have you considered doing any regression analysis? Statistical analyses are too basic.

Yes, regression analyses have been done. However, based on the results obtained so far, we have decided that they would be the subject of a separate study.

The discussion talks about students as if they were competitive athletes and I understand that this is not the case.

The surveyed students (female and male) had high levels of physical activity due to, among other things, their study profile (Tourism and Recreation) and some of them were members of university sports teams.

We would like to emphasize that very few publications have addressed this topic in the population of university students.

Also, the discussion is confusing and short. Strengths are not strengths. Just because there are many variables does not mean that the statistical analysis is adequate.

The Discussion section was revised and new information, including research on university students, was added.

Holistic health includes the social domain as well.

Thank you for this valuable input. We fully agree.

The conclusions are too brief. The reviewer.

The Conclusions section was expanded.

Reviewer 2 Report

Dear Authors,
This is a study investigating the impact of the epidemic on the physical and mental health of college students. This is a meaningful investigation. However, I suggest improvements to the manuscript that would improve reader convenience.

1. Abstract
a) Supplementary note, the name of the tool used for the questionnaire analysis.
b) Add proper nouns related to the theme of the special issue, delete "women; men"

2. Introduction
a) Line 47, "More than 28 million...". Are you referring to the loss of 28 million lives? Or the loss of life? Please determine the way and grammar of the sentence.
b) The subject of this manuscript is college students. However, the author's description of the impact of the epidemic on college students has only three lines in lines 103-105. However other content talks about lines 40-102. This ratio is inappropriate. It is recommended to reduce the content of lines 58-80 to expand the influence of college students in the epidemic or corroborate relevant literature.
c) The blocking policy can cite the management cases of other countries or regions, and the manuscript will be more objectively stated.
Example: DOI: 10.3390/ijerph18115717

3.Materials and Methods
a) Figure 1, does this picture have the right to remove the image? It is recommended that the author use other software graphics to present, or indicate whether the source of this picture can be used.
b) Supplement the experimental flow chart so that readers will have a clearer understanding of the process of this manuscript.

4. Results
a) Regarding the well-being assessment, how did the authors determine that there was a significant difference between before and after?
b) Please add a note below Table 2, Table 3, Table 4, and Table 5, p < 0.001 *

5. Discussion
It is good to cite the literature to confirm the experimental results. However, the author's presentation process should state the experimental results from a personal standpoint and deduce the cause of the results. rather than repeated citations.

6. Conclusions
The conclusion statement is pure.

Author Response

Dear Reviewer,

You will find our response in attachment.

Best regards,

Robert Podstawski

International journal of EnvironmentaL Research and Public Health

14 November 2022

            Below you will find our responses to the review of the article entitled “The influence of COVID-19 on university students’ well-being, physical activity, body composition, and strength endurance”. Manuscript ID: ijerph-2004320

Above all, we would like to thank the Reviewers for their valuable contributions to preparing the paper for publication. The Reviewers’ comments have been taken into consideration in the revision process, and changes in the manuscript are highlighted in yellow. A detailed description of the changes is presented below in Table 1.

Table 1. Specific comments and responses

Responses to the comments made by Reviewer 2

Dear Authors,

This is a study investigating the impact of the epidemic on the physical and mental health of college students. This is a meaningful investigation. However, I suggest improvements to the manuscript that would improve reader convenience.

Thank you for recognizing the significance of our research. The Reviewer's comments were taken into consideration in the revision process.

1. Abstract

a) Supplementary note, the name of the tool used for the questionnaire analysis.

b) Add proper nouns related to the theme of the special issue, delete "women; men"

Both issues were resolved in the revised manuscript.

2. Introduction

a) Line 47, "More than 28 million...". Are you referring to the loss of 28 million lives? Or the loss of life? Please determine the way and grammar of the sentence.

b) The subject of this manuscript is college students. However, the author's description of the impact of the epidemic on college students has only three lines in lines 103-105. However other content talks about lines 40-102. This ratio is inappropriate. It is recommended to reduce the content of lines 58-80 to expand the influence of college students in the epidemic or corroborate relevant literature.

c) The blocking policy can cite the management cases of other countries or regions, and the manuscript will be more objectively stated. Example: DOI: 10.3390/ijerph18115717

Thank you for this comments.

a)      We are referring to excess years of life lost, which is an alternative measure of premature mortality due to COVID-19. This term is correct and widely used, including in scientific writing.

b)       Additional information about college/university students (CUS) was provided. This issue was also raised by Reviewer 1, but the impact of the pandemic on CUS has been rarely researched, and the relevant information is scarce.

c)      Thank you for this suggestion. The recommended reference was used in the Introduction section.

3.Materials and Methods

a) Figure 1, does this picture have the right to remove the image? It is recommended that the author use other software graphics to present, or indicate whether the source of this picture can be used.

b) Supplement the experimental flow chart so that readers will have a clearer understanding of the process of this manuscript.

a)      The person shown in the photographs is Anna Maria Podstawska, who is one of the co-authors and who has given written consent to use her image in the paper. The signed consent form has been forwarded to the Editor.

b)      The flow chart was modified for improved readability.

4. Results

a) Regarding the well-being assessment, how did the authors determine that there was a significant difference between before and after?

b) Please add a note below Table 2, Table 3, Table 4, and Table 5, p < 0.001 *

a)      Significant differences were determined at p <0.05. The significance of differences between the number of points obtained before and during the pandemic was assessed. The testing procedure is described in the Statistical Analysis subsection (2.4).

b)      The relevant notes were added under the tables.

5. Discussion It is good to cite the literature to confirm the experimental results. However, the author's presentation process should state the experimental results from a personal standpoint and deduce the cause of the results. rather than repeated citations.

The discussion was improved by presenting the experimental results from a personal point of view and identifying potential factors that influenced the results.

6. Conclusions The conclusion statement is pure.

The Conclusions section was revised.

General comments

The revised manuscript contains 68 references.

The affiliations of Krzysztof Borysławski and Anna Maria Podstawska have changed.

We would like to thank both Reviewers for extensive and thorough reviews which have helped us to improve the quality of this manuscript. We hope that the revised manuscript is suitable for publication. We are always open to constructive criticisms, and should the Reviewers have any additional comments, we ask that they provide us with specific suggestions. Once again, thank you for the time devoted to reviewing our manuscript.

Yours sincerely,

Robert Podstawski, on behalf of all co-authors.

Round 2

Reviewer 1 Report

Dear authors,

I think the methodology is still unclear. When I say that reliability data of the scales should be included, I am referring for example to sensitivity and specificity.

Furthermore, it is not clear to me that your sample is high-performance athletes.

I am happy to contribute suggestions to your study.

The reviewer.

Author Response

Below you will find our responses to the 2nd round review of the article entitled “The influence of COVID-19 on university students’ well-being, physical activity, body composition, and strength endurance”. Manuscript ID: ijerph-2004320

Once again we would like to thank the Reviewers for their valuable contributions to preparing the paper for publication. The Reviewers’ comments have been taken into consideration in the revision process, and changes in the manuscript are highlighted in yellow. A detailed description of the changes is presented below in Table 1.

On behalf of the research team, I would like to thank the Reviewers for a prompt and thorough review of our manuscript. Their comments have provided us with a new perspective on the discussed problem and have enabled us to make substantive editorial improvements in the manuscript. We hope that the revised manuscript will meet your expectations. Once again, thank you for the time devoted to reviewing our manuscript.

Yours sincerely,

Robert Podstawski, on behalf of all co-authors.

Table 1. Specific comments and responses

Comment

Response

Responses to the comments made by Reviewer 1

I think the methodology is still unclear. When I say that reliability data of the scales should be included, I am referring for example to sensitivity and specificity.

As suggested by the Reviewer, information on sensitivity and specificity was provided in the following sentence: The POMS questionnaire is a valid and reliable tool characterized by overall detection rate 80% with a sensitivity of 55% and a specificity of 84% [33], for measuring negative mood states, such as psychological distress, as well as positive mood states, such as vigor [34], including in conjunction with PA [35].

Furthermore, it is not clear to me that your sample is high-performance athletes.

We would like to kindly inform you that the students taking part in the study are not high-performance athletes. We have checked again the characteristics of the respondents in the "Participants" subsection and there is no information about this there. Since the assessment of the level of physical activity showed that the respondents (men and women) had a high level of physical activity (>1500 MET), the discussion also included studies addressing the impact of pandemic covid 19 on the well-being of athletes. To clarify this point, one of the sentences in the discussion has been completed to read as follows: The absence of organized training sessions was an additional stressor for athletes, and a similar phenomenon also occurred among tourism and physical recreation students who participate in sports activities, which may have been due to the fact that they were characterised by high levels of physical activity [51].

I am happy to contribute suggestions to your study.

We are very grateful for your indicated comments

Reviewer 2 Report

Dear authors,
I think the quality of this revised manuscript has improved. In my opinion, the author should reconfirm the details of the abstract and the narrative style, grammar, etc. of the conclusion. It can reach the level recommended to the editor-in-chief to include this manuscript.
good luck,

Author Response

Dear Reviewer,

Thank you very much for your comments.

19 November 2022

Below you will find our responses to the 2nd round review of the article entitled “The influence of COVID-19 on university students’ well-being, physical activity, body composition, and strength endurance”. Manuscript ID: ijerph-2004320

Once again we would like to thank the Reviewers for their valuable contributions to preparing the paper for publication. The Reviewers’ comments have been taken into consideration in the revision process, and changes in the manuscript are highlighted in yellow. A detailed description of the changes is presented below in Table 1.

On behalf of the research team, I would like to thank the Reviewers for a prompt and thorough review of our manuscript. Their comments have provided us with a new perspective on the discussed problem and have enabled us to make substantive editorial improvements in the manuscript. We hope that the revised manuscript will meet your expectations. Once again, thank you for the time devoted to reviewing our manuscript.

Yours sincerely,

Robert Podstawski, on behalf of all co-authors.

Table 1. Specific comments and responses

Comment

Response

Responses to the comments made by Reviewer 2

I think the quality of this revised manuscript has improved. In my opinion, the author should reconfirm the details of the abstract and the narrative style, grammar, etc. of the conclusion. It can reach the level recommended to the editor-in-chief to include this manuscript.
good luck,

As noted by the Reviewer, the abstract and conlusions have once again been checked and corrected by a native speaker. We ran a "grammar-check" using the Word (software) editor. Without changes, it scored 90% and 93% for both the abstract and conclusions sections, respectively. We did some "minor revisions" (see highlighted yellow text in the attachment) to increase the scores (for both) to 98%. 
